# Selective and Efficient Catalytic Oxygenation of Alkyl Aromatics Employing H₂O₂ Catalyzed by Simple Porphyrin Iron(II) under Mild Conditions

Xin-Yan Zhou [1], Bin He [2], Yu Zhang [1], Jia-Ye Ni [1], Qiu-Ping Liu [1], Mei Wang [1], Hai-Min Shen [1,*] and Yuan-Bin She [1,*]

[1] College of Chemical Engineering, Zhejiang University of Technology, Hangzhou 310014, China; 17816876326@163.com (X.-Y.Z.); lym15665861061@163.com (Y.Z.); njy970615@163.com (J.-Y.N.); liuqiuping@zjut.edu.cn (Q.-P.L.); zxy19960312@yeah.net (M.W.)
[2] Fuyang Branch of Hangzhou Municipal Ecology and Environment Bureau, Fuyang District, Hangzhou 311400, China; sia2016_zhaohui@126.com
* Correspondence: haimshen@zjut.edu.cn (H.-M.S.); sheyb@zjut.edu.cn (Y.-B.S.); Tel.: +86-159-8846-0002 (H.-M.S.)

**Abstract:** The excessive utilization of additives in chemical reactions is a troublesome problem in industrial processes, due to their adverse effects on equipment and processes. To acquire oxidative functionalization of alkyl aromatics under additive-free and mild conditions, a large library of metalloporphyrins was applied to the oxygenation of alkyl aromatics as catalysts with H₂O₂ as an oxidant. On the basis of systematic investigation of the catalytic performance of metalloporphyrins, it was discovered that, surprisingly, only porphyrin irons(II) possessed the ability to catalyze the oxygenation of alkyl aromatics with H₂O₂ under additive-free conditions and with satisfying substrate scope. Especially with 5,10,15,20-tetrakis(2,6-dichlorophenyl) porphyrin iron(II) (T(2,6-diCl)PPFe) as the catalyst, the substrate conversion reached up to 27%, with the selectivity of 85% to the aromatic ketone in the representative oxygenation of ethylbenzene with H₂O₂ as oxidant and without any additive used. The study of apparent kinetics and mechanisms in the optimal oxygenation system was also conducted in detail. Based on thorough exploration and characterization, the source of the superior catalytic performance of T(2,6-diCl)PPFe was acquired mainly as its planar structure, the low positive charge in the metal center, and better solubility in the oxygenation mixture, which favored the approach of reactants to the catalytic center, and the interaction between the metal center and H₂O₂. The beneficial interaction between T(2,6-diCl)PPFe and H₂O₂ was verified through cyclic voltammetry measurements and UV–vis absorption spectra. In comparison to previous studies, in this work, an efficient, selective, and additive-free means was developed for the oxygenation of alkyl aromatics under mild conditions, which could act as a representative example and a valuable reference for industrial processes in oxygenation of alkyl aromatics, and a great advance in the realization of oxygenation of alkyl aromatics under additive-free and mild conditions.

**Keywords:** metalloporphyrins; oxygenation; alkyl aromatics; hydrogen peroxide; aromatic ketone

## 1. Introduction

Selective oxygenation of petrochemical products is a fundamental change in the chemical industry, because of its maximum atom economy, shortest transformation path, and low impact on the environment [1–3]. One of these reactions is the oxygenation of alkyl aromatics, which results in the forming of a variety of significant and valuable oxygen-containing molecules, such as ketones, aldehydes, alcohols, acids, and their derivatives. For instance, ethylbenzene is a commercially important chemical material in industrial chemical syntheses, as well as a typical compound of secondary benzylic C—H intermediate that can be oxidized to acetophenone. Acetophenone is a common industrial intermediate synthesized

by the Friedel-Crafts acylation procedure. The process has low reactivity and causes severe equipment corrosion. Due to the inertness of C—H bonds, harsh conditions are frequently required, such as high temperatures, powerful oxidants, and caustic additions, resulting in poor selectivity, deep oxygenation, and increased environmental costs [4–6]. Efficient and selective oxygenation of ethylbenzene to acetophenone has attracted much research attention recently.

To achieve a satisfactory selectivity towards partially oxygenated products, transition-metal complex catalysts and ecologically acceptable oxidants, such as molecular oxygen ($O_2$) [7,8], *tert*-butyl hydroperoxide (TBHP) [9,10], and hydrogen peroxide ($H_2O_2$), were encouraged. Because using $O_2$ requires high temperature and pressure conditions and using TBHP produces huge amounts of liquid waste, $H_2O_2$ is used more frequently. For instance, Bryliakov and co-workers employed chiral *bis*-amine-*bis*-pyridine manganese complexes to catalyze the enantioselective hydroxylation of ethylbenzene and its derivates with $H_2O_2$, and the yield of 1-Phenylethanol reached up to 50% [11]. Zhao and co-workers carried out a bimetal catalyst with the metal center of Co and Ni reaching up to 80% ethylbenzene conversion with the oxidant of TBHP [12]. Ha and co-workers reported W-MnSBA-15 mesoporous catalysts with great catalytic activity and recyclability yielding 86% ethylbenzene conversion [13]. As for another oxidant oxygen, Nie and colleagues reported benzylic C—H bond oxygenation employing nitrogen-doped carbon as a catalyst in the presence of tert-butyl hydroperoxide, the obtained conversion of ethylbenzene was 99% with a selectivity of 93% towards aromatic ketone [14]. Song and co-workers reported ethylbenzene and its derivates oxygenation using acetonitrile as the solvent rather than caustic acetic acid employing copper phosphate and N-hydroxyphthalimide under $O_2$ conditions. When ethylbenzene was utilized as a substrate, the yield of acetophenone reached up to 93% [15]. In this work, 2 new symmetrical and unsymmetrical ligands were connected to the diiron(III) metal center, and proved to have great catalytic activity towards alkene and alkane (58% ethylbenzene conversion and 77% selectivity towards ketones) [16]. Almost all of the above-mentioned works used additives in the reaction, which were corrosive in the chemical industry. In earlier work, we examined systematically the catalytic system, using metalloporphyrins and molecular oxygen ($O_2$) to oxidize aromatic benzylic C—H bonds towards corresponding ketones and alcohols [17,18]. The alkyl aromatics' oxygenation could be greatly sped up with effective catalytic systems, and the efficient approach was achieved using heterogeneous catalytic systems. It was impossible to ignore the fact that using heterogeneous catalysts resulted in a larger catalyst loading and a loss in catalytic efficiency, which makes it more difficult to separate the catalysts, and uses more energy to recover unconverted substrates.

Metalloporphyrins, as the representative compounds of Cytochrome P450, have proven to be an ideal candidate for C—H bonds oxygenation because of the efficiently active sites and environmental friendliness. Although a vast library of porphyrin-based homogeneous and heterogeneous catalysts for oxygenation of aromatic benzylic C—H bonds have been extensively studied, the development of an effective metalloporphyrin catalyst for oxygenation of aromatic hydrocarbons to ketones that achieves high selectivity and satisfying conversion under mild conditions, is still a work in progress. The catalytic activity of metalloporphyrins, like that of native enzymes, is linked to structure, which can be altered by central metal and porphyrin ligands, to satisfy the demands of various catalytic reactivity. Inspired by our previous work employing simple metalloporphyrins, we investigated the oxygenation of alkyl aromatics employing metalloporphyrins as catalysts and $H_2O_2$ as an oxidant systematically. The work is an efficient and selective additive-free oxygenation of aromatic benzylic C—H bonds using $H_2O_2$ as an oxidant catalyzed by simple metalloporphyrins (T(2,6-diCl)PPFe) under non-additive and mild conditions, achieving better ethylbenzene conversion (up to 27.44%) and improved ketone selectivity (up to 85%). Cyclic voltammetry measurements and UV-vis absorbance spectra were used to demonstrate the interaction between T(2,6-diCl)PPFe and $H_2O_2$. This study is not only a reference in the area of the relationship between the catalytic performance of

alkyl aromatics and the structures of metalloporphyrins, but also a significant advancement for the widespread use of metalloporphyrins as the catalyst.

## 2. Experimental Section

Chemicals, materials, characterization, instrumentation, and synthesis of metalloporphyrin, are all covered in the Supplementary Materials section.

### 2.1. Syntheses of Metalloporphyrins

Metalloporphyrins used in this study were synthesized using the standard process described in the literature [19–22].

The following is a typical process for porphyrin ligands: to begin, 150 mmol of substituted benzaldehyde was dissolved in propionic acid (550 mL). The solution was then progressively injected with redistilled pyrrole (150 mmol) while it was heated to reflux under a nitrogen atmosphere. The solution was then maintained, stirring and refluxing, for another 2.0 h. After the solution had cooled to room temperature, 800 mL of methanol was added to it. When methanol was added, a large amount of precipitate formed, which was collected using filtration. The precipitate was washed 3 times with 100 mL methanol, until the filtrate was clear. Finally, crude products were purified using a silica column with cyclohexane and dichloromethane.

The following is a typical process for metalloporphyrins: porphyrin ligand (0.20 mmol) and metal acetate (2.0 mmol) were dissolved using 100 mL DMF, and heated to reflux under a nitrogen atmosphere. After 24.0 h, DMF was evaporated using rotary evaporation. The solid residue was then dissolved in dichloromethane (60 mL), and washed 4 times with water (4 × 150 mL). The crude products were further refined using a silica column with cyclohexane and dichloromethane. Before usage, the metalloporphyrins were vacuum-dried at 80 °C for 8.0 h. The Supplementary Materials section contains details on the synthesis and characterization of porphyrins and metalloporphyrins.

### 2.2. Autoxidation of Ethylbenzene

Autoxidation of alkyl aromatics with $H_2O_2$ was studied using ethylbenzene as a model. In the normal working procedure, a reaction tube (35 mL) was filled with ethylbenzene (0.0106 g, 0.1 mmol) and $H_2O_2$ (0.8 mmol). The reaction tube was then stirred while being heated to the designed reaction temperature. After 6.0 h, the resulting mixture and 1 mL of naphthalene solution (internal standard) were transferred to a volumetric flask and diluted to 5 mL with acetonitrile. Finally, GC and HPLC were used to examine the conversion and selectivity of alkyl aromatics autoxidation. By comparing it to a standard sample, the product was recognized qualitatively.

### 2.3. Catalytic Oxygenation of Ethylbenzene and Its Derivates

Initially, the catalytic efficiency of the metalloporphyrins was evaluated for ethylbenzene oxygenation in acetonitrile at 70 °C, and the optimal conditions were applied to the other substrates. In a 35 mL reaction tube, 0.01 mmol of catalyst, 0.1 mmol of the substrate, 0.8 mmol of $H_2O_2$, and $CH_3CN$ to complete 1 mL of volume, were sequentially combined and stirred under nitrogen atmosphere, in a thermostated bath. After 6.0 h, the resulting mixture and 1 mL of naphthalene solution (internal standard) were transferred to a volumetric flask and diluted to 5 mL with acetonitrile. Finally, GC and HPLC were used to examine the conversion and selectivity of ethylbenzene and its derivates oxygenation.

### 2.4. Kinetic Study

To investigate the catalytic characteristics of different metalloporphyrins, apparent kinetic studies of ethylbenzene oxygenation were conducted under 50 °C, 60 °C, and 70 °C, using TPPFe, T(4-Br)PPFe, and T(2,6-diCl)PPFe as catalysts. The autoxidation of ethylbenzene was conducted under 50 °C, 60 °C, and 70 °C. A reaction tube was filled with solvent, catalyst, and substrates in the standard procedure. The reaction mixture

was immediately cooled to 25 °C after stirring for 1.0, 1.5, 2.0, 2.5, and 3.0 h, under the set condition. Finally, the resulting mixture and 1 mL of naphthalene solution (internal standard) were accurately diluted to 5 mL with acetonitrile. The resulting solution was injected into gas chromatography, to measure conversion and selectivity.

### 2.5. Products Analyses

The oxygenation products were examined using a Thermo Trace 1300 gas chromatographer equipped with a Flame Ionization Detector and a TG-5MS capillary column (30 m × 0.32 mm × 0.25 μm). A Thermo Ultimate 3000 HPLC chromatographer with Photodiode Array Detector and Amethyst C18-H liquid chromatography column (250 mm × 4.6 mm × 0.25 μm) was used to examine aromatic carboxylic acids. For quantification, an internal calibration technique was applied, in which calibration curves for all substrates and products were used to establish the respective response factor. The internal standard for GC analysis was naphthalene, while the internal standard for HPLC analyses was 2-naphthalene carboxylic acid. The samples were tested before and after the addition of triphenylphosphine, which transforms the hydroperoxide to the corresponding alcohol, to ensure a valid measurement of the alkyl hydroperoxides.

## 3. Results and Discussion

### 3.1. Characterizations

In addition to NMR and ESI-MS characterizations, UV-vis spectra were obtained in order to confirm the structure of metalloporphyrins and their ligands (Figure 1). The successful synthesis of metalloporphyrins was revealed by the absorption peak at 420 nm [23–26]. And the variations in the chemical structure and solubility of metalloporphyrins in DMF were primarily responsible for the variations in the absorption intensity. Cyclic voltammetry in tetrabutylammonium hexafluorophosphate (TBAPF$_6$) solution (0.025 mol/L in DMF) was used to assess the performance of metalloporphyrins in electron transfer. At the proper voltage, all of the metalloporphyrins under investigation showed clear oxidation potentials and reduction potentials, which suggested that they had potential in electron transfer to serve as catalysts in oxygenation reactions (Figure 2). Thermogravimetric analyses (TGA) of matching metalloporphyrins were carried out under a nitrogen environment from 25 °C to 800 °C, to examine the thermal stability. For almost all metalloporphyrins, no noticeable mass loss occurred before 250 °C (Figure 3). The metalloporphyrins used in this study were sufficiently stable in the catalytic environment.

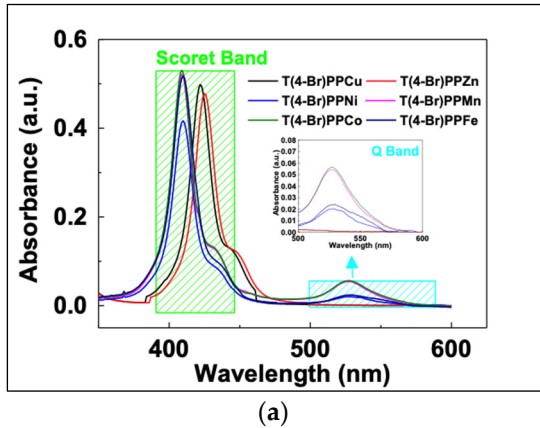
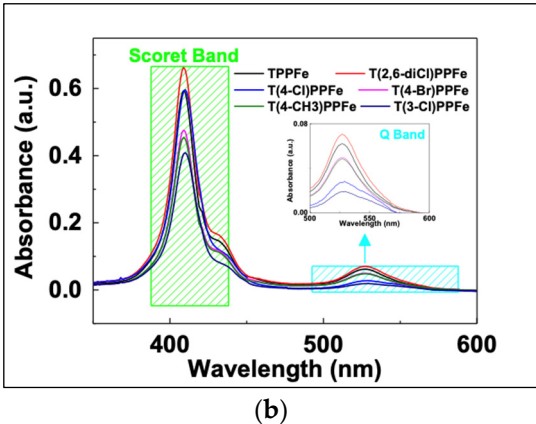

**Figure 1.** UV-Vis absorption spectra of metalloporphyrins (**a**,**b**).

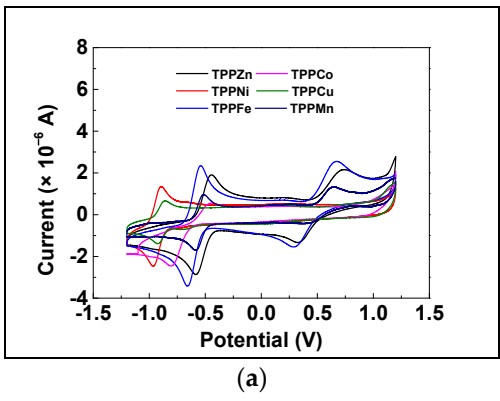 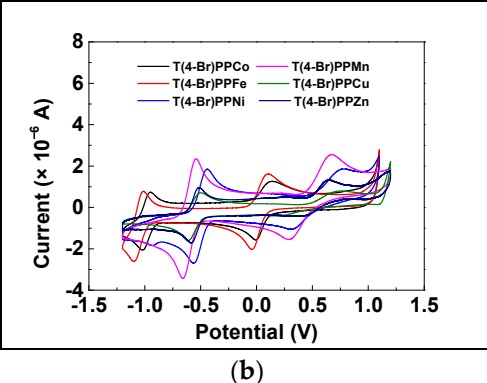

**Figure 2.** Cyclic voltammetric curves for metalloporphyrins (**a**,**b**).

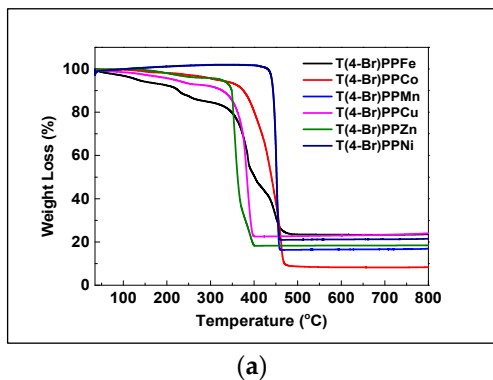 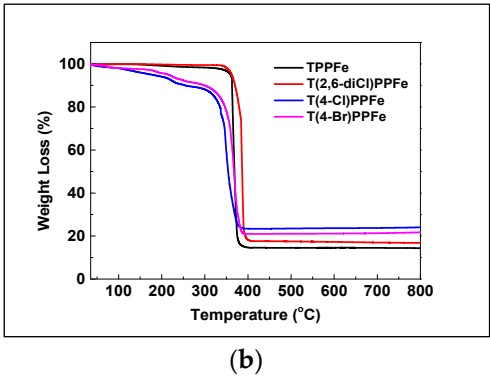

**Figure 3.** Thermogravimetric analysis of metalloporphyrins (**a**,**b**).

### 3.2. Preliminary Exploratory Research

Some early studies were carried out before conducting a full study on the oxygenation of C—H bonds. The primary issue was establishing the proper reaction temperature. Two catalysts, tetrakis(4-Bromine-phenyl) porphyrin iron(II) (T(4-Br)PPFe) and tetrakis(phenyl) porphyrin iron(II) (TPPFe), were investigated, in ethylbenzene and cumene oxygenation. Ketones and alcohols were found in the reaction. Because deep oxygenation occurred at higher temperatures, oxygenation temperature was an important factor to consider. The catalytic activity of metalloporphyrins was studied at temperatures ranging from 50 °C to 70 °C, and the results are listed in Tables 1 and S1. Under autoxidation conditions, no evidence of oxygenated compounds was discovered when the oxygenation temperature was less than 70 °C. The catalytic activity of metalloporphyrins reached the best results at 70 °C, with a conversion of 19.71%. TPPFe had a ketone selectivity of up to 70%, and T(4-Br)PPFe had a ketone selectivity of 93% (Entry 10 and Entry 15 in Table 1). When triphenylphosphine was added to the resulting combination, there was no triphenylphosphine oxide found, indicating that all hydroperoxide had been transformed to the equivalent alcohol (Entry 16 and Entry 17 in Table 1). Table 1 also indicated that the metal center was primarily responsible for the catalytic activity of metalloporphyrins in catalyzed oxygenation processes, and the porphyrin ligand lacked evident catalytic activity (Entry 18 and Entry 19 in Table 1). When the salts of Fe(II) were used as catalysts (Entry 20), no oxygenated products were formed due to their poor dispersion, which was also strong proof of the great catalytic performance of metalloporphyrins to some degree. Because greater yields were obtained for all catalysts at 70 °C and no deep oxygenation products were formed, oxygenation of ethylbenzene to corresponding ketones at 70 °C was a viable option. In this study, the reaction temperature was set to 70 °C.

**Table 1.** Preliminary exploration on alkyl aromatics oxygenation temperature employing ethylbenzene as model substrate [a].

| Entry | Catalysts | Temperature (°C) | Conversion (%) | Selectivity (%) | | | |
|---|---|---|---|---|---|---|---|
| | | | | $R_1$=O | $R_1$-OH | $R_1$-OOH | $R_2$-COOH |
| 1 | - | 50 | <1% | - | - | - | - |
| 2 | - | 55 | <1% | - | - | - | - |
| 3 | - | 60 | <1% | - | - | - | - |
| 4 | - | 65 | <1% | - | - | - | - |
| 5 | - | 70 | <1% | - | - | - | - |
| 6 | TPPFe | 50 | 12.50 | 74 | 26 | - | - |
| 7 | TPPFe | 55 | 15.35 | 76 | 24 | - | - |
| 8 | TPPFe | 60 | 17.15 | 77 | 23 | - | - |
| 9 | TPPFe | 65 | 19.46 | 78 | 22 | - | - |
| 10 | TPPFe | 70 | 19.71 | 79 | 21 | - | - |
| 11 | T(4-Br)PPFe | 50 | 9.26 | 77 | 23 | - | - |
| 12 | T(4-Br)PPFe | 55 | 12.08 | 80 | 20 | - | - |
| 13 | T(4-Br)PPFe | 60 | 12.83 | 83 | 17 | - | - |
| 14 | T(4-Br)PPFe | 65 | 13.66 | 87 | 13 | - | - |
| 15 | T(4-Br)PPFe | 70 | 14.35 | 93 | 7 | - | - |
| 16 | TPPFe [b] | 70 | 19.52 | 79 | 21 | - | - |
| 17 | T(4-Br)PPFe [b] | 70 | 14.25 | 92 | 8 | - | - |
| 18 | TPP | 70 | <1% | - | - | - | - |
| 19 | T(4-Br)PP | 70 | <1% | - | - | - | - |
| 20 | Fe(OAc)$_2$ | 70 | <1% | - | - | - | - |

[a] Reaction tube (35 mL), ethylbenzene (0.1 mmol, 0.0106 g), $H_2O_2$ (0.8 mmol), 6.0 h, 550 rpm. [b] Sample was analyzed after the addition of triphenylphosphine.

### 3.3. Effect of Central Metal on Catalytic Oxygenation of Alkyl Aromatics

The central metal has a significant influence on the catalytic performance of simple metalloporphyrin, which is the main source of metalloporphyrin catalyst. With TPP and T(4-Br)PP as porphyrin ligands and 6 kinds of transition metals as representative central metals, the effect of metal centers on the catalytic oxygenation of ethylbenzene was investigated. Tables 2, S2 and S3 revealed that only metalloporphyrins iron(II) exceeded all other central metal complexes in terms of catalytic performance. The cyclic voltammetry curves of several metalloporphyrins and UV–vis spectroscopy of the metalloporphyrins with $H_2O_2$ were gathered to answer the question of why only metalloporphyrins iron(II) can catalyze substrate efficiently. As shown in Figure 4, both oxygenation and reduction currents developed, indicating that the corresponding metalloporphyrins transported electrons smoothly, especially when forming high-valence Fe-oxo complexes and activating the substrate. Cyclic voltammetry curves results revealed that TPPFe and T(4-Br)PPFe had the lowest oxygenation potential and superior electron transport skills, which allowed electron transfer to occur more quickly, resulting in high conversion.

**Table 2.** Preliminary exploration on the effect of metal centers on alkyl aromatics oxygenation employing ethylbenzene as model substrate [a].

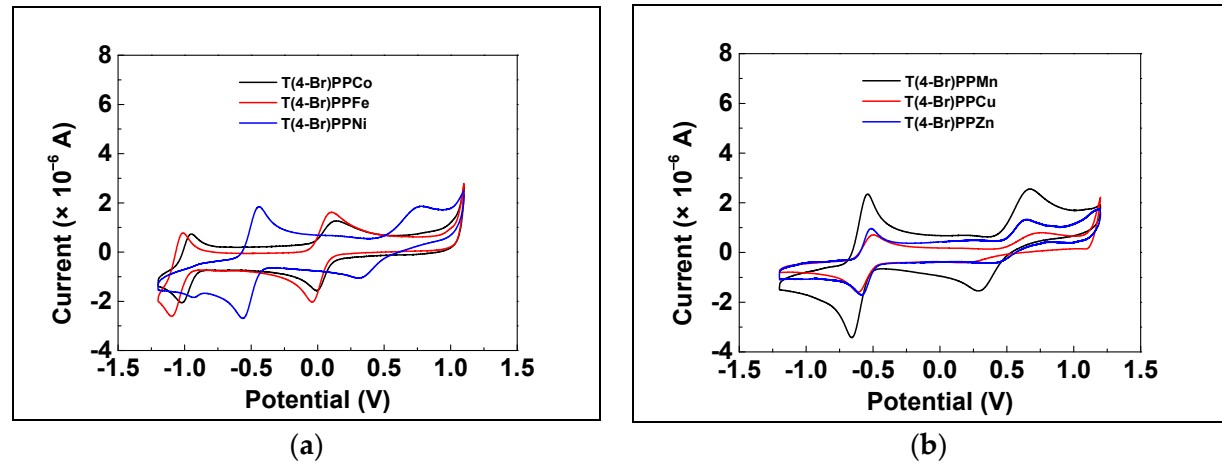

| Entry | Catalysts | Conversion (%) | Selectivity (%) | | | |
|-------|-----------|----------------|-----------------|------|--------|---------|
| | | | $R_1$=O | $R_1$-OH | $R_1$-OOH | $R_2$-COOH |
| 1 | TPPFe | 19.71 | 79 | 21 | - | - |
| 2 | TPPCo | <1% | - | - | - | - |
| 3 | TPPMn | <1% | - | - | - | - |
| 4 | TPPNi | <1% | - | - | - | - |
| 5 | TPPCu | <1% | - | - | - | - |
| 6 | TPPZn | <1% | - | - | - | - |
| 7 | T(4-Br)PPFe | 14.35 | 93 | 7 | - | - |
| 8 | T(4-Br)PPCo | <1% | - | - | - | - |
| 9 | T(4-Br)PPMn | <1% | - | - | - | - |
| 10 | T(4-Br)PPNi | <1% | - | - | - | - |
| 11 | T(4-Br)PPCu | <1% | - | - | - | - |
| 12 | T(4-Br)PPZn | <1% | - | - | - | - |

[a] Reaction tube (35 mL), ethylbenzene (0.1 mmol, 0.0106 g), $H_2O_2$ (0.8 mmol), metalloporphyrins (10%, mol/mol), 70 °C, 6.0 h, 550 rpm.

**Figure 4.** Cyclic voltammetric curves for metalloporphyrins (**a**,**b**).

UV–vis spectroscopy was used to monitor the metalloporphyrin reaction with $H_2O_2$. The spectrum (Figures 5 and S1) initially showed a signal band at 420 nm, but when the experiments were carried out using $H_2O_2$, a new band at 430 nm was found. The active species in the reaction, Fe(III)OOH intermediates, had a characteristic band in the range of 430 nm, and the presence of PorFe(III)OOH was confirmed by ESI-MS, indicating that the radical mechanism was valid. When compared to the reaction of metalloporphyrins iron(II) and $H_2O_2$, the reaction of metalloporphyrins Co(II) and Mn(II) displayed fewer spectrum changes, indicating lower catalytic activity of Co(II) and Mn(II). Metalloporphyrins Cu(II), Zn(II), and Ni(II), showed no spectrum change when exposed to $H_2O_2$.

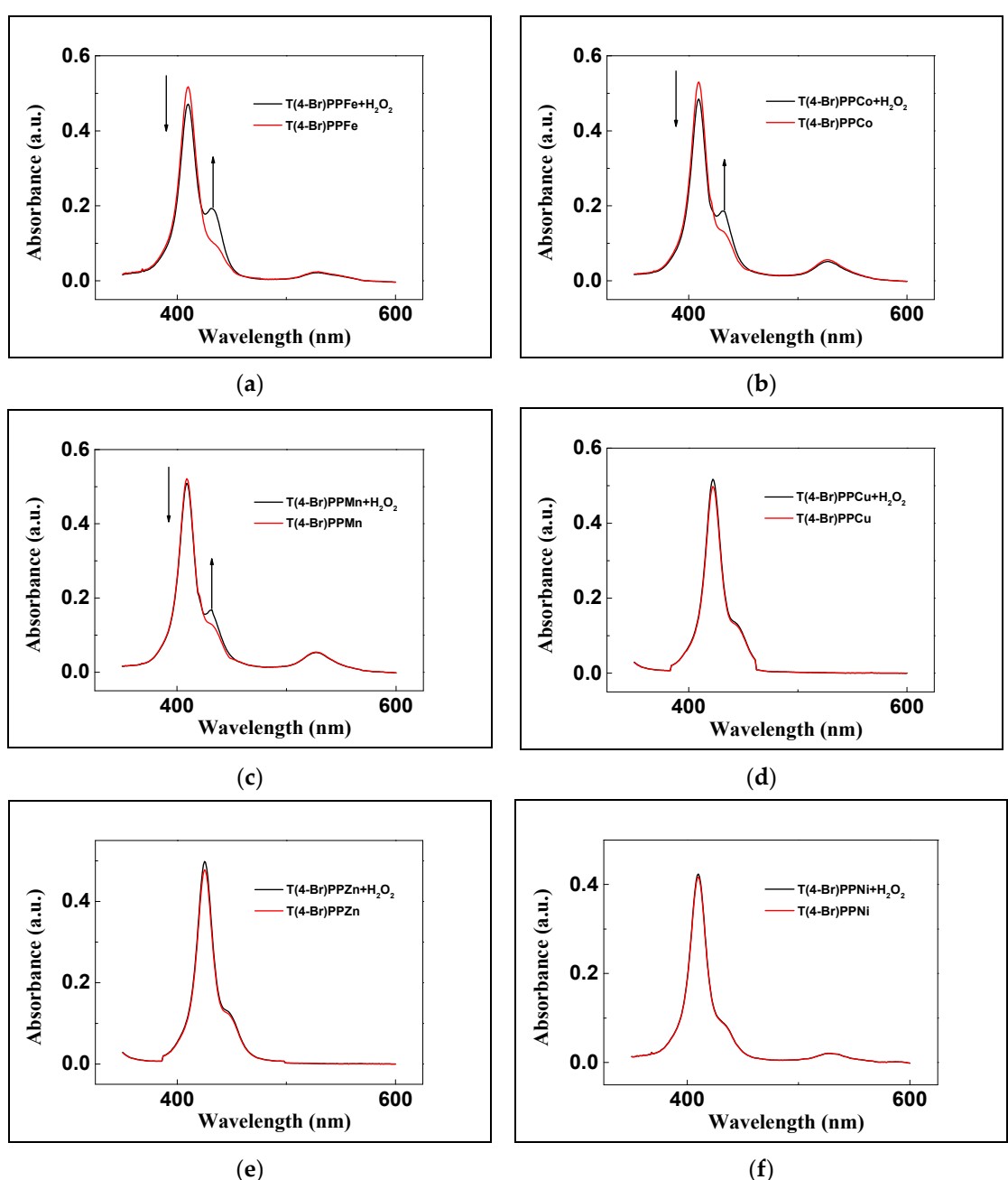

**Figure 5.** UV-Vis absorption spectra of T(4-Br)PPFe (**a**); T(4-Br)PPCo (**b**); T(4-Br)PPMn (**c**); T(4-Br)PPCu (**d**); T(4-Br)PPZn (**e**); T(4-Br)PPNi (**f**) with $H_2O_2$.

### 3.4. Effect of Porphyrin Ligands on Alkyl Aromatics Catalytic Oxygenation

Because ligand structures affected the catalytic performance of metalloporphyrins, a series of metalloporphyrins iron(II) with various substituent groups was investigated, employing ethylbenzene and cumene as a model substrate. Tables 3, S4 and S5, showed that all of the metalloporphyrins iron(II) used had a satisfactory catalytic performance on the transformation of both ethylbenzene and cumene to acetophenone, and the metalloporphyrins iron(II) with more notable steric hindrance substituent groups resulted in higher yields. T(2,6-diCl)PPFe was the most efficient, with a conversion rate of 27.44% and a selective of 85% towards acetophenone. The greater yields of T(2,6-diCl)PPFe could be attributed to its high acetonitrile solubility and plane structure, as shown in Figure S8. To evaluate the solubility, a series of metalloporphyrins iron(II) was dissolved in 10 mL acetonitrile. Figure S4 shows that the solution was colorless for T(2-Cl)PPFe, T(2-CH$_3$)PPFe,

and T(4-OCH$_3$)PPFe, which had no catalytic activity, and purple-red for the others, indicating that the difference in conversion may be attributable to solubility. The central metal of metalloporphyrins was therefore protected from attack by radical species in the reaction, and electron transfer capacity was improved, resulting in increased conversion and selectivity.

**Table 3.** Effect of porphyrin structures on oxygenation of alkyl aromatics employing ethylbenzene as model substrate [a].

| Entry | Catalysts | Conversion (%) | Selectivity (%) | | | |
|---|---|---|---|---|---|---|
| | | R$_1$=O | R$_1$=O | R$_1$-OH | R$_1$-OOH | R$_2$-COOH |
| 1 | TPPFe | 19.71 | 79 | 21 | - | - |
| 2 | T(2-Cl)PPFe | <1% | - | - | - | - |
| 3 | T(3-Cl)PPFe | 17.46 | 86 | 14 | - | - |
| 4 | T(4-Cl)PPFe | 20.84 | 82 | 18 | - | - |
| 5 | T(2,6-diCl)PPFe | 27.44 | 85 | 15 | - | - |
| 6 | T(2-CH$_3$)PPFe | <1% | - | - | - | - |
| 7 | T(3-CH$_3$)PPFe | 13.87 | 82 | 18 | - | - |
| 8 | T(4-CH$_3$)PPFe | 20.52 | 78 | 22 | - | - |
| 9 | T(3-OCH$_3$)PPFe | 12.89 | 83 | 17 | - | - |
| 10 | T(4-OCH$_3$)PPFe | <1% | - | - | - | - |
| 11 | T(3-F-4-Br)PPFe | 13.63 | 87 | 13 | - | - |
| 12 | T(2-F-4-Br)PPFe | 12.57 | 85 | 15 | - | - |
| 13 | T(2-Cl-4-Br)PPFe | 14.12 | 87 | 13 | - | - |
| 14 | T(3-Cl-4-Br)PPFe | 14.21 | 86 | 14 | - | - |
| 15 | T(2,3,6-triCl)PPFe | 3.87 | 85 | 15 | - | - |
| 16 | T(2,3,5-triCl)PPFe | 3.76 | 82 | 18 | - | - |
| 17 | T(2,3,6-triF)PPFe | 1.62 | 82 | 18 | - | - |
| 18 | T(2,3,5-triF)PPFe | 2.01 | 83 | 17 | - | - |

[a] Reaction tube (35 mL), ethylbenzene (0.1 mmol, 0.0106 g), H$_2$O$_2$ (0.8 mmol), metalloporphyrins (10%, mol/mol), 70 °C, 6.0 h, 550 rpm.

### 3.5. Effect of Catalyst Loading and Oxidant Amount

The resulting catalytic system in ethylbenzene oxygenation was further investigated in terms of oxidant amount and catalyst loading to increase aromatic ketones conversion. Tables 4, S6 and S7 show when the catalyst loading was increased from 5% to 25%, the conversion results remained nearly unchanged. Tables 5, S8 and S9 show that as the oxidant amount increased from 4:1 (mol/mol) to 8:1 (mol/mol) catalyzed by T(2,6-triCl)PPFe, the conversion of ethylbenzene increased from 10.11% to 27.44%. The selectivity towards ketone was not improved by further increasing the oxidant amount which resulted in the formation of acid. Increases in the amount of oxidant, produced a decrease in selectivity rather than a rise in conversion. To summarize, using T(2,6-diCl)PPFe as the catalyst, an efficient catalytic system for direct and selective oxygenation of ethylbenzene and its substrates was developed, with the conversion of ethylbenzene reaching 27.44% and selectivity of 85% towards aromatic ketone.

**Table 4.** Effect of catalyst amount on alkyl aromatics oxygenation employing ethylbenzene as model substrate [a].

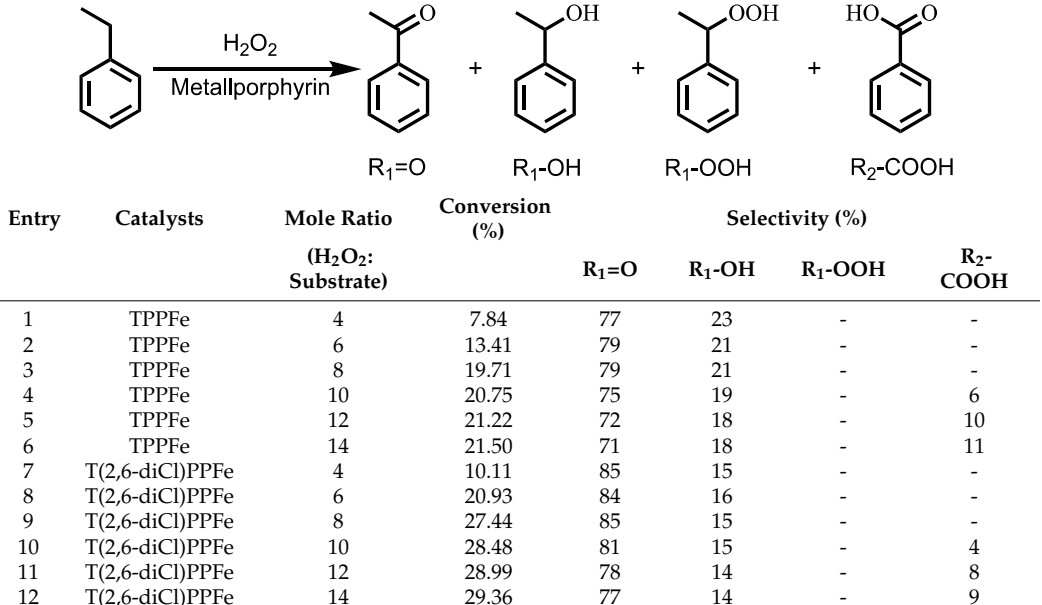

| Entry | Catalysts | Catalyst Amount (%, mol/mol) | Conversion (%) | Selectivity (%) | | | |
|---|---|---|---|---|---|---|---|
| | | | | $R_1$=O | $R_1$-OH | $R_1$-OOH | $R_2$-COOH |
| 1 | TPPFe | 0.005 | 18.95 | 80 | 20 | - | - |
| 2 | TPPFe | 0.010 | 19.71 | 79 | 21 | - | - |
| 3 | TPPFe | 0.015 | 20.53 | 81 | 19 | - | - |
| 4 | TPPFe | 0.020 | 20.72 | 81 | 19 | - | - |
| 5 | TPPFe | 0.025 | 21.09 | 80 | 20 | - | - |
| 6 | T(2,6-diCl)PPFe | 0.005 | 26.70 | 85 | 15 | - | - |
| 7 | T(2,6-diCl)PPFe | 0.010 | 27.44 | 85 | 15 | - | - |
| 8 | T(2,6-diCl)PPFe | 0.015 | 27.61 | 83 | 17 | - | - |
| 9 | T(2,6-diCl)PPFe | 0.020 | 27.83 | 84 | 16 | - | - |
| 10 | T(2,6-diCl)PPFe | 0.025 | 27.94 | 86 | 14 | - | - |

[a] Reaction tube (35 mL), ethylbenzene (0.1 mmol, 0.0106 g), $H_2O_2$ (0.8 mmol), metalloporphyrins, 70 °C, 6.0 h, 550 rpm.

**Table 5.** Effect of oxidant amount on alkyl aromatics oxygenation employing ethylbenzene as model substrate [a].

| Entry | Catalysts | Mole Ratio ($H_2O_2$: Substrate) | Conversion (%) | Selectivity (%) | | | |
|---|---|---|---|---|---|---|---|
| | | | | $R_1$=O | $R_1$-OH | $R_1$-OOH | $R_2$-COOH |
| 1 | TPPFe | 4 | 7.84 | 77 | 23 | - | - |
| 2 | TPPFe | 6 | 13.41 | 79 | 21 | - | - |
| 3 | TPPFe | 8 | 19.71 | 79 | 21 | - | - |
| 4 | TPPFe | 10 | 20.75 | 75 | 19 | - | 6 |
| 5 | TPPFe | 12 | 21.22 | 72 | 18 | - | 10 |
| 6 | TPPFe | 14 | 21.50 | 71 | 18 | - | 11 |
| 7 | T(2,6-diCl)PPFe | 4 | 10.11 | 85 | 15 | - | - |
| 8 | T(2,6-diCl)PPFe | 6 | 20.93 | 84 | 16 | - | - |
| 9 | T(2,6-diCl)PPFe | 8 | 27.44 | 85 | 15 | - | - |
| 10 | T(2,6-diCl)PPFe | 10 | 28.48 | 81 | 15 | - | 4 |
| 11 | T(2,6-diCl)PPFe | 12 | 28.99 | 78 | 14 | - | 8 |
| 12 | T(2,6-diCl)PPFe | 14 | 29.36 | 77 | 14 | - | 9 |

[a] Reaction tube (35 mL), ethylbenzene (0.1 mmol, 0.0106 g), $H_2O_2$, metalloporphyrins (10%, mol/mol), 70 °C, 6.0 h, 550 rpm.

### 3.6. Kinetic Study

Noticeably, we have optimized a catalytic system for selective catalytic oxygenation of aromatic benzylic C—H bonds employing $H_2O_2$. A series of kinetic experiments were conducted from 50 °C to 70 °C, with ethylbenzene serving as a substrate, 3 kinds of representative metalloporphyrins were employed as catalysts, and 3 models were employed to measure the apparent activation energies (Ea). From the results summarized in Figure 6 and Table 6, the pseudo-first-order kinetic model (Figure 6) had the highest correlation coefficients when fitting the reaction data of substrate autoxidation and catalytic oxygenation with $H_2O_2$. The apparent activation energies (Ea) were calculated using the Arrhenius equation: lnk = −(Ea/R) × (1/T) + lnk$_0$, and declined in the following order: T(4-Br)PPFe

(57.09 kJ/mol) > TPPFe (52.10 kJ/mol) > T(2,6-diCl)PPFe (41.09 kJ/mol). In the instance of ethylbenzene oxygenation, the link between Ea and catalytic activity is clear: lower apparent activation energy resulted in higher ethylbenzene conversion and selectivity towards acetophenone.

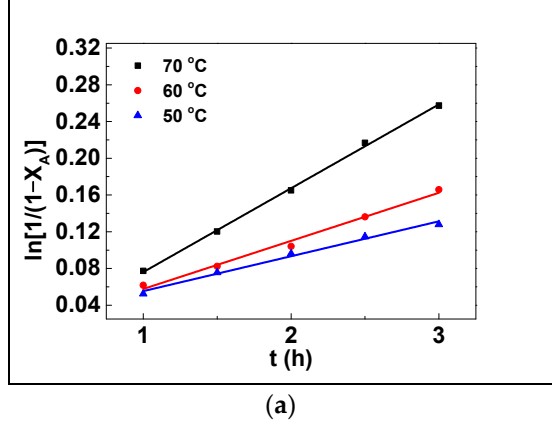 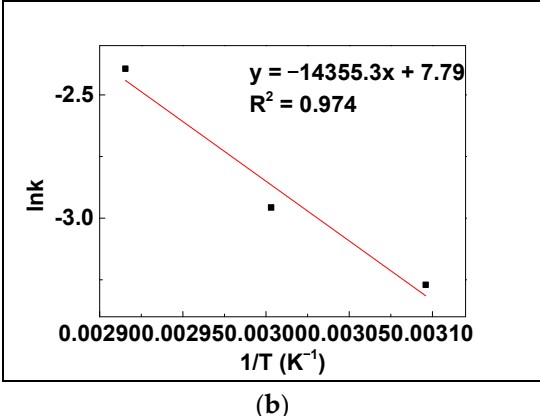

(**a**)                  (**b**)

**Figure 6.** Pseudo-first-order fits for oxygenation of ethylbenzene with $H_2O_2$ catalyzed by T(2,6-diCl)PPFe (**a**,**b**).

**Table 6.** The pseudo-first-order kinetic parameters in ethylbenzene catalytic oxygenation.

| Entry | Catalysts | Temp. (°C) | k (L·mol$^{-1}$·h$^{-1}$) | $R^2$ | Average Intercepts | Ea (kJ/mol) |
|---|---|---|---|---|---|---|
| 1 | TPPFe | 50 | 0.0217 | 0.9920 | 0.0060 | 51.37 |
| 2 | | 60 | 0.0306 | 0.9941 | | |
| 3 | | 70 | 0.0665 | 0.9916 | | |
| 4 | T(4-Br)PPFe | 50 | 0.0117 | 0.9929 | −0.0044 | 53.31 |
| 5 | | 60 | 0.0170 | 0.9936 | | |
| 6 | | 70 | 0.0374 | 0.9964 | | |
| 7 | T(2,6-diCl)PPFe | 50 | 0.0380 | 0.9906 | 0.0025 | 40.24 |
| 8 | | 60 | 0.0524 | 0.9906 | | |
| 9 | | 70 | 0.0913 | 0.9988 | | |

### 3.7. Mechanism on the Reaction Pathways

To examine whether the mechanism of metalloporphyrin iron(II) catalytic oxygenation of ethylbenzene and its derivates is a free radical process, we employed a mechanism test. Radical inhibitor bromotrichloromethane ($CBrCl_3$), 2-bromo-2-methylpropane (($CH_3)_3CBr$), and diphenylamine ($Ph_2NH$) were added into the reaction tubes, and the conversion was decreased from 20.87%, to 3.12%, 3.79%, and 3.60% respectively, thus suggesting a free radical process. When $CBrCl_3$ was employed to catch radicals, GC-MS analyses revealed (1-bromoethyl)benzene and (1-chloroethyl)benzene, indicating the presence of benzyl radical ($C_8H_9$·). When ($CH_3)_3CBr$ was employed to catch radicals, GC-MS analyses revealed both (1-bromoethyl)benzene and *tert*-Butanol, indicating the presence of hydroxyl radical (HO·) (Figure S6). Furthermore, the radical species in the process of ethylbenzene oxygenation were examined using an electron paramagnetic resonance (EPR) study. In the reaction mixture, ethyl phenyl radical ($C_8H_9$·) as well as ethyl phenyl peroxy radical ($C_8H_9OO$·), and ethyl phenyl oxidative radical ($C_8H_9O$·) were found when the spin trap was utilized (Figure 7). The detected radical intermediates were primarily ethyl phenyl ($C_8H_9$·), hydroxyl (HO·), ethyl phenyl peroxy radical ($C_8H_9OO$·), and ethyl phenyl oxygen radical ($C_8H_9O$·).

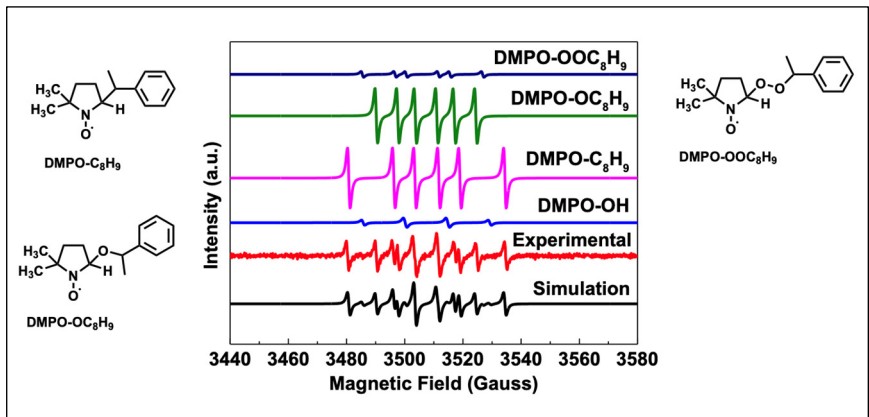

**Figure 7.** Computed and experimental EPR spectrum of DMPO spinning adducts and DMPO spinning adducts structures.

For oxygenation promoted by a series of iron complexes, the "water-assisted mechanism" [27,28] was extensively accepted. As shown in Scheme 1, a "water-assisted mechanism" for the oxygenation of ethylbenzene and its derivates by metalloporphyrin Fe(II) can be described as follows. The initiation of the catalytic reaction is the metalloporphyrins iron(II) oxidized to the PorFe(III)OOH (A), which was detected by ESI-MS. The PorFe(III)OOH (A) converted to a reactive species Fe (III)-oxo complex (B) with $H_2O$. Followed by this step, the Fe (III)-oxo complex (B) extracts a hydrogen atom from the $CH_2$ group of ethylbenzene and its derivates to form benzyl radical and short-lived electron-deficient radical intermediate (C). Then, the OH group rebounded to benzyl radical and formed the corresponding alcohol, which would be further oxidized to the ketone. The ESI-MS and UV/Vis spectroscopic were collected to confirm the presence of PorFe(III)OOH.

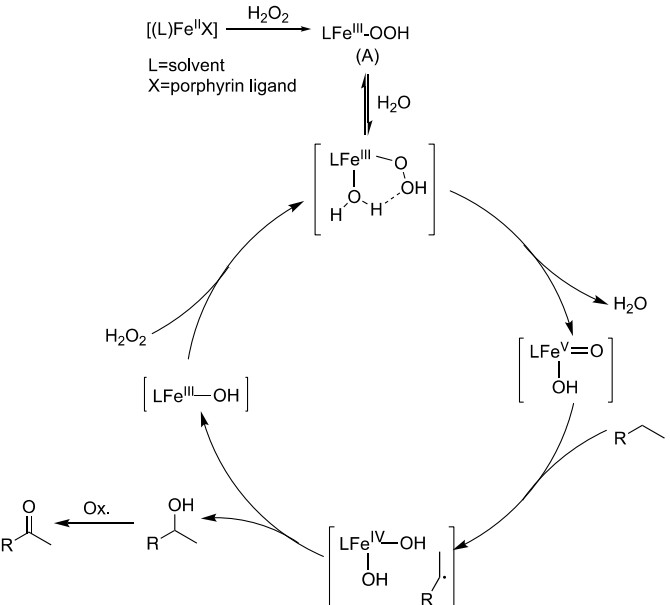

**Scheme 1.** Proposed mechanism of alkyl aromatics oxygenation employing $H_2O_2$ catalyzed by metalloporphyrins [29].

### 3.8. Comparison with Different $H_2O_2$ Catalytic Systems and Substrate Scope

The scope of the substrate was expanded, from ethylbenzene to several ethylbenzene derivates having secondary C—H bonds, when the optimal method for the direct oxygenation of ethylbenzene and its derivates to ketones was established. To evaluate the substrate adaptability of the catalytic system in this work, the substrate was expanded to various

alkyl aromatics. All ethylbenzene and its derivates tested could be oxidated to ketones with a selectivity of 50% to 90%, as shown in Table 7. Due to the high stability of 1-ethyl-2-nitrobenzene, the oxygenation of 1-ethyl-2-nitrobenzene resulted in a conversion of 14.73% and a selectivity towards secondary benzylic ketone of 50%. With a selectivity of about 80% except for 1-ethyl-2-nitrobenzene, all substrates yielded ketones and alcohols, but no acid. The method reported in this study was a high-potential and practicable solution for converting ethylbenzene and its derivates to valuable ketones directly under additive-free conditions using $H_2O_2$ catalyzed by T(2,6-diCl)PPFe. Due to the great catalytic activity of metalloporphyrins Fe(II), the oxygenation system obtained from the study may be employed in conventional alkyl aromatics oxygenation. The method described in this paper was compared to other references of secondary benzylic C—H bond oxygenation systems, as shown in Table 8. In this paper, we developed a catalytic system that used simple metalloporphyrins and mild conditions to achieve high selectivity. We kept the conversion under control so that only useful ketones and alcohols were formed. Meanwhile, among six different metals, we studied the underlying reasons why only metalloporphyrin Fe(II) could oxygenate aromatic C—H bonds.

**Table 7.** Ethylbenzene and its derivates oxygenation employing $H_2O_2$ catalyzed by T(2,6-diCl)PPFe [a].

| Entry | Substrates | Conversion (%) | Selectivity (%) | | | |
|---|---|---|---|---|---|---|
| | | | $R_1$=O | $R_1$-OH | $R_1$-OOH | $R_2$-COOH |
| 1 | ethylbenzene | 27.44 | 85 | 15 | - | - |
| 2 | 1-ethyl-2-nitrobenzene | 14.73 | 50 | 50 | - | - |
| 3 | 1-ethyl-2-bromobenzene | 26.71 | 81 | 19 | - | - |
| 4 | diphenylmethane | 25.28 | 83 | 17 | - | - |
| 5 | fluorene | 50.49 | 90 | 10 | - | - |
| 6 | tetralin | 65.67 | 76 | 24 | - | - |
| 7 | indane | 73.78 | 90 | 10 | - | - |
| 8 | 1-ethyl-4-nitrobenzene | 36.30 | 88 | 12 | - | - |

**Table 7.** *Cont.*

| Entry | Substrates | Conversion (%) | Selectivity (%) | | | |
|---|---|---|---|---|---|---|
| | | | R$_1$=O | R$_1$-OH | R$_1$-OOH | R$_2$-COOH |
| 9 | H$_3$CO—⬡—C$_2$H$_5$ | 38.26 | 84 | 16 | - | - |
| 10 | Br—⬡—C$_2$H$_5$ | 40.65 | 80 | 20 | - | - |

[a] Reaction tube (35 mL), substrate (0.1 mmol), H$_2$O$_2$ (0.8 mmol), metalloporphyrins (10%, mol/mol), 70 °C, 550 rpm.

**Table 8.** Comparison with some reported systems for oxidative transformation of ethylbenzene and its derivates to ketones with H$_2$O$_2$.

| Entry | Main Products | Conditions | Conversion (%) | Selectivity (%) | Ref. |
|---|---|---|---|---|---|
| 1 | acetophenone | supported iron (0.3%, m/m), H$_2$O$_2$ (3.5 equiv), n-Bu$_4$NBr, 1,4-dioxane, 70 °C, 24.0 h | 98 | 91 | [30] |
| 2 | acetophenone | nanocomposite CoFe$_2$O$_4$@SiO$_2$@MIL-53(Fe) (1.2%, m/m), H$_2$O$_2$ (2.3 equiv), H$_2$O, 25 °C, 4.0 h | 94 | 99 | [31] |
| 3 | acetophenone | Mn(II) complex (0.1%, mol/mol), H$_2$O$_2$ (5 equiv), H$_2$O, AcOH (10 equiv), 70 °C, 10.0 h | 89 | 91 | [32] |
| 4 | acetophenone | Cu(II) complex (0.1%, m/m), H$_2$O$_2$ (1 equiv), CH$_3$CN, 70 °C, 8.0 h | 47 | 98.0 | [33] |
| 5 | acetophenone | copper nanoparticle in Al$_2$O$_3$ supported Co(II) and Cu(II) complex (10%, m/m), H$_2$O$_2$ (1.25 equiv), CH$_3$CN, 50 °C, 4.0 h | 95 | 95.0 | [34] |
| 6 | acetophenone | V(IV) and V(V) complex (1%, m/m), H$_2$O$_2$ (2.0 equiv), CH$_3$CN, 80 °C, 20.0 h | 65 | 82 | [35] |

**Table 8.** *Cont.*

| Entry | Main Products | Conditions | Conversion (%) | Selectivity (%) | Ref. |
|---|---|---|---|---|---|
| 7 | | Tetra-Fe (III) cluster (0.04%, m/m), H$_2$O$_2$ (2.5 equiv), AcOH (0.5 equiv), 32 °C, 3.0 h | 60 | 76 | [36] |
| 8 | | Phen-MCM-41 (Co-Catalyst) (2.5%, m/m), H$_2$O$_2$ (2.0 equiv), NHPI, CH$_3$CN, 80 °C, 4.5 h | 78 | 76 | [37] |
| 9 | | Pd/PdO/Fe$_3$O$_4$@PGQD (0.4%, m/m), H$_2$O$_2$ (2.0 equiv), MeOH, 25 °C, 20.0 h | 77 | 99 | [38] |
| 10 | | CuFe$_2$O$_4$ nanoparticles (0.02%, m/m), H$_2$O$_2$ (3.0 equiv), CH$_3$CN, 60 °C, 24.0 h | 56 | 89 | [39] |
| 11 | | Fe(II) complex (0.1%, m/m), H$_2$O$_2$ (0.2 equiv), CH$_3$CN, 25 °C, 1.5 h | 80 | 56 | [40] |
| 12 | | VO@g-C$_3$N$_4$ (10.0%, m/m), H$_2$O$_2$ (1.5 equiv), CH$_3$CN, 25 °C, 12.0 h | 99 | 99 | [41] |
| 13 | | nanospheres of magnetite (Fe$_3$O$_4$@m-SiO$_2$) support FeSi/Ag/VO nanocomposite (5.0%, m/m), H$_2$O$_2$ (1.5 equiv), CH$_3$CN, 60 °C, 8.0 h | 46 | 72 | [42] |
| 14 | | Mn(II) complex (0.5%, m/m), H$_2$O$_2$ (3.5 equiv), AcOH (14 equiv), 0 °C, 0.5 h | 72 | 93 | [43] |
| 15 | | heterogeneous organocatalyst glycoluril (5.0%, m/m), H$_2$O$_2$ (1.2 equiv), H$_2$O, 60 °C, 3.0 h | 99 | 98 | [44] |
| 16 | | iron-anchored naphthyl-azo catalyst (PS-Fe-NAPA) (1.0%, m/m), H$_2$O$_2$ (2.5 equiv), CH$_3$CN, 60 °C, 7.0 h | 96 | 93 | [45] |
| 17 | | oxygen bridged homobinuclear Mn(II) compounds (1.0%, m/m), H$_2$O$_2$ (2.5 equiv), CH$_3$CN, 60 °C, 7.0 h | 79 | 99 | [46] |

**Table 8.** *Cont.*

| Entry | Main Products | Conditions | Conversion (%) | Selectivity (%) | Ref. |
|---|---|---|---|---|---|
| 18 |  | iron based catalyst [(PDP)Fe(OTf)$_2$] (1.0%, m/m), H$_2$O$_2$ (4 equiv), EHA (10 equiv), 0 °C, 2.5 h | 19 | 95 | [47] |
| 19 |  | metalloporphyrins T(2,6-diCl)PPFe (10.0%, m/m), H$_2$O$_2$ (8.0 equiv), CH$_3$CN, 70 °C, 12.0 h | 27 | 85 | This work |
| 20 |  | metalloporphyrins T(2,6-diCl)PPFe (10.0%, m/m), H$_2$O$_2$ (8.0 equiv), CH$_3$CN, 70 °C, 12.0 h | 36 | 88 | This work |

## 4. Conclusions

Using ethylbenzene and its derivatives as a model substrate, an efficient and selective additive-free alkyl aromatics oxygenation, employing H$_2$O$_2$ as oxidant and simple metalloporphyrin iron(II) as catalyst, was accomplished. The main benefit of this catalytic system was that it maintained a balance between improved selectivity, milder conditions, and higher conversion, owing to the great electron transport capacity of metalloporphyrin iron(II), and the environmentally friendly oxidant H$_2$O$_2$. Meanwhile, this research found the relationship between the central metal of metalloporphyrins and catalytic activity in aromatic benzylic C—H bonds oxygenation employing H$_2$O$_2$. The cyclic voltammetry curves of metalloporphyrins indicated better electron transferring performances of metalloporphyrin iron(II), which determined the catalytic reactivity of the complexes. Compared with current references, the optimum balance between better selectivity, milder conditions, and simpler catalysts, was proved to be the key advantage of our work. As a result, our strategy for converting alkyl aromatics with optimized, T(2,6-diCl)PPFe and H$_2$O$_2$ was an effective, manageable, and appealing way to functionalize ethylbenzene and its derivates. It was also a crucial example of using simple metalloporphyrins to functionalize and use alkyl aromatics. We anticipate more research into the aromatic benzylic C—H bonds oxygenation using easily prepared metalloporphyrins under additive-free and mild conditions.

**Supplementary Materials:** The following supporting information can be downloaded at: https://www.mdpi.com/article/10.3390/pr11041187/s1.

**Author Contributions:** X.-Y.Z.: Data curation; Formal analysis; Methodology; Software; Visualization; Roles/Writing-original draft; Writing-review and editing. B.H.: Data curation; Formal analysis; Methodology; Software; Visualization. Y.Z.: Data curation; Writing-review and editing. J.-Y.N.: Data curation; Writing-review and editing. Q.-P.L.: Data curation; Writing-review and editing. M.W.: Data curation; Writing-review and editing. H.-M.S.: Conceptualization; Funding acquisition; Investigation; Project administration; Roles/Writing-original draft; Writing-review and editing; Validation; Resources. Y.-B.S.: Conceptualization; Funding acquisition; Project administration; Resources. All authors have read and agreed to the published version of the manuscript.

**Funding:** This research was funded by the National Natural Science Foundation of China (Grant No. 21878275, 22178322, 22138011, 21776259).

**Data Availability Statement:** Not applicable.

**Conflicts of Interest:** The authors declare no conflict of interest.

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
