# Peer review of "Selective and Efficient Catalytic Oxygenation of Alkyl Aromatics Employing H2O2 Catalyzed by Simple Porphyrin Iron(II) under Mild Conditions"

_processes, doi:10.3390/pr11041187_

Round 1

Reviewer 1 Report

In this manuscript the oxygenation of ethylbenzene using porphyrin iron(II) as catalyst in the presence of H2O2 was studied. The effect of metal type and ligand type of the catalyst on the catalyst activity, kinetic and mechanism of the reaction were investigated. This manuscript can be considered to publish.

In section 3.3 the higher activity of Fe rather than the other central metals is explained on the basis of the cyclic volumetric curves and UV-Vis absorption spectra of the different catalysts as shown in figures 4, 5. These curves are very similar for Fe and Co as central metals. The oxygenation potential in cyclic volumetric curves and the changes in the UV-Vis spectra for both metals are similar. With considering these observations, how is it explained the high difference between these metals' catalytic activities?

Reviewer 2 Report

This manuscript provides an extensive description of a series of porphyrin complexes with first-row transition metals and their subsequent use of catalysts for the oxygenation with H2O2 of aromatic molecules containing aliphatic groups. The authors find that only the iron complexes are effective catalysts for this transformation. This is not surprising as iron porphyrin complexes are well-known catalysts for oxidation reactions. In the next step, they study a series of iron complexes modifying the porphyrin ligand with different steric and electronic properties. However, it seems like the main cause of the different reactivity results from the solubility of the iron complex in acetonitrile. They optimize the catalysis for the best catalyst and, then, study a range of ethylbenzene derivatives in the reaction. The mechanistic studies with radical traps and EPR measurements are interesting and support a radical mechanism. The authors have a lot of data on both characterization of the complexes and the catalytic studies. However, the manuscript is written more like a thesis chapter than a research article. Many ideas are repeated throughout the manuscript and are even found in the same paragraph multiple times. As an example of this in section 3.4 the same idea is repeated up to three times in one paragraph Line 267: “greater yields of T(2,6-diCl)PPFe could be attributed to its high acetonitrile solubility and plan structures”

Line 272-273: “The great catalytic performance of T(2,6-272 diCl)PPFe was most likely due to the bulky substituent groups and plan structures, which”

Line 275: “Steric hindrance was given by the bulky substituent 275 groups, which resulted in planar structures,”

Another example of this repetitive writing style is found in Section 3.6. kinetic study. In particular in the last part of the paragraph (lines 319-325), where the same idea is repeated in also the same words in three consecutive sentences:

“the link between Ea and catalytic activity is clear: lower apparent activation energy resulted in higher ethylbenzene conversion and selectivity towards acetophenone. The different catalytic performances between various metalloporphyrins were mostly due to the different capacity to decrease the apparent activation energy in ethylbenzene oxygenation. Thus, differences in apparent activation energies (Ea) represent catalytic activity and can explain the catalytic performances of various metalloporphyrins in ethylbenzene oxygenation.”

The three sentences convey the exact same meaning lower Ea leads to higher conversion and selectivity. In general, the manuscript needs to be rewritten in a more concise and succinct manner. Moreover, a lot of the figures in Figure 6 (section 3.6. kinetic study) should be moved to the supporting information. A representative example of such as Fig 6a and Fig 6b can be included in the main text while the rest can go to the supporting information.

Moreover, section 3.6. kinetic study is confusing as it is not clear in the figures what the difference is between the pseudo-first order and pseudo-second order fits as the same things are represented in the x- and y-axis. This should be cleared and better explained.

The English of the manuscript should be revised as in some cases it makes the reading of the manuscript difficult.

I cannot accept the manuscript in its current form. It would greatly benefit from a complete rewrite that is both clearer and more concise, as well as a thorough check of the English used throughout the document.

Round 2

Reviewer 2 Report

The authors have greatly improved the readability and clarity of the manuscript.